# Immunotherapy and Hepatocellular Cancer: Where Are We Now?

**DOI:** 10.3390/cancers14184523

**Published:** 2022-09-19

**Authors:** Marine Valery, Baptiste Cervantes, Ramy Samaha, Maximiliano Gelli, Cristina Smolenschi, Alina Fuerea, Lambros Tselikas, Caroline Klotz-Prieux, Antoine Hollebecque, Valérie Boige, Michel Ducreux

**Affiliations:** 1Département de Médecine Oncologique, Gustave Roussy, F-94805 Villejuif, France; 2Département d’Anesthésie, Chirurgie et Interventionnel, Gustave Roussy, F-94805 Villejuif, France; 3Département d’Innovation Thérapeutique, Gustave Roussy, F-94805 Villejuif, France; 4Inserm Unité Dynamique des Cellules Tumorales, Gustave Roussy, Université Paris-Saclay, F-94805 Villejuif, France

**Keywords:** hepatocellular carcinoma, liver cancer, immunotherapy, immune checkpoint inhibitors, antiangiogenic treatments, tyrosine kinase inhibitors

## Abstract

**Simple Summary:**

HCC is the third-leading cause of cancer death worldwide. Prior to the immunotherapy era, treatment of advanced HCC was dominated by antiangiogenic tyrosine kinase inhibitors, with a quite poor prognosis with a median survival time of around one year. In this review, we aimed to elaborate on all the advances in immunotherapy in HCC from the indisputable first-line therapy atezolizumab–bevacizumab in advanced HCC to promising new strategies. We highlight the new standard of care in advanced HCC, namely, the combination of immune checkpoint inhibitors (ICI) durvalumab–tremelimumab, and also discuss the different combinations of antiangiogenic drugs with ICI. We develop how the combination of ICI with locoregional therapies could become a future treatment in early and intermediate-stage HCC, and, finally, we outline several new cell-therapy-based strategies to follow in the coming years.

**Abstract:**

Immunotherapy has demonstrated its effectiveness in many cancers. In hepatocellular carcinoma (HCC), promising results shown in the first phase II studies evaluating anti-PD-1 or anti-PD-L1 monotherapies resulted in their approval in the United States. Approval was not obtained in Europe; subsequent randomized studies in first- or second-line treatment did not confirm these initial results. However, first data with immunotherapy plus antiangiogenic treatments or dual immunotherapy combinations were positive. In this context, the combination of bevacizumab and atezolizumab took the lead. The IMbrave150 trial revealed an improved objective response rate (ORR), progression-free survival, and overall survival with this combination versus the previous standard, sorafenib. Subsequent results of dual immunotherapy with the anti-CTLA-4 and anti-PD-1 monotherapies tremelimumab and durvalumab (also superior to sorafenib monotherapy) confirmed the value of using a combination in first-line treatment. These significant therapeutic advances, and the increase in ORR, raise two main questions. Whereas response was very limited with previous treatments, the ORR reported with these new combinations are between 20% and 30%. This raises the question of whether immunotherapy (ICI single agent, combination of ICI with antiangiogenic agent or other antitumoral treatment) can be used in patients beyond those in BCLC group C, the traditional candidate group for systemic therapy. We have thus seen an increasing number of patients previously treated with trans-arterial chemoembolization (BCLC group B) receiving these new treatments, and we develop the results of several studies combining loco-regional therapies and immunotherapy-based systemic treatments. The other major question is that of how and when to use these medical treatments as “adjuvants” to interventional radiology or surgery; the results of several works are discussed for this purpose. In this review, we cover all of these points in a fairly comprehensive manner.

## 1. Introduction

The liver has an essential function in maintaining biological homeostasis. Receiving the portal blood, it must deal with exogenous antigens from ingested food or various microorganisms. Avoiding autoimmune phenomena involves diverse immunotolerance mechanisms consisting of innate and acquired immune system cells and immunosuppressive cytokines (such as IL-10, TGFβ, and “indoleamine 2,3-dioxygenase”) [1,2,3]. In patients with hepatocellular carcinoma (HCC) with underlying chronic cirrhosis, persistent inflammation leads to PD-1/PD-L1 overexpression and upregulation of other immune checkpoint molecules (CTLA-4, LAG3, and TIM3), resulting in cytotoxic CD8+ T-cell depletion and CD4+ Foxp3 Treg cell recruitment. HCC develops in an immune-exhausted tumor microenvironment (TME).

Globally, there are two types of HCC regarding the TME. The so-called “hot” tumors, which represent only 20–25% of HCCs, with a predominantly TCD8+ lymphocyte infiltrate, which overexpress PD-1/PD-L1 and CTLA-4, two main targets of immunotherapies [4,5]. These HCCs are therefore good candidates for treatment with checkpoint inhibitors (PD-1/PD-L1 and CTLA-4 inhibitors), whose main role, which is now well known, is to allow a lifting of the inhibition of the lymphocyte immune response to the tumor cell. The second type of tumors are immunologically “cold” tumors, where the immune microenvironment is richer in immunosuppressive cells such as Foxp3+ regulatory T-cells and poorer in effector TCD8+ [4,5]. One of the levers to make the tumor more sensitive to immunotherapy is the use of antiangiogenic agents. Indeed, several antiangiogenic drugs are already validated in the treatment of advanced HCC (e.g., sorafenib, lenvatinib, cabozantinib, regorafenib, etc. [6]). HCC tumor growth relies heavily on the neoangiogenesis pathway involving growth factors such as vascular endothelial growth factor A (VEGFA) and platelet-derived growth factor (PDGF). Among other functions, VEGFA promotes the recruitment of Treg cells and inhibits that of effector TCD8+ [7,8]. It also induces cell death preferentially of TCD8+ and not of Treg, favoring an immunosuppressed TME [7,8]. On this basis, normalization of the vascularization and downregulation of neoangiogenesis pathways through the use of antiangiogenic drugs has become a main strategy to enhance the efficacy of immunotherapy drugs on HCC [9].

Regarding the use of immunotherapy in combination with local regional treatments (local destruction, chemoembolization, radioembolization, stereotactic radiotherapy), the rationale is based on the release of tumor neo-antigens by the local treatment, thus initiating an immune response with the recruitment of immune cells. Thus, the tumor is “sensitized” to the immunotherapy, with TME made more favorable.

## 2. Immune Checkpoint Inhibitor (ICI) Monotherapy in Advanced HCC: Successes and Failures

The demonstration of the efficacy of PD-1 and PD-L1 inhibitor monotherapy in many indications has prompted a similar development in HCC. Nivolumab and pembrolizumab monotherapies received FDA (Food and Drug Administration) approval in the treatment of HCC following the early phase CheckMate 40 and KEYNOTE-224 studies, respectively [10,11], which reported objective response rates (ORR) of around 20%. However, the two phase III trials, CheckMate 459, which evaluated nivolumab versus sorafenib in first-line treatment, and KEYNOTE-240, which compared pembrolizumab versus placebo after first-line treatment with sorafenib, both failed to meet their primary endpoints [12,13]. In CheckMate 459, median overall survival (mOS) was 16.4 months with nivolumab and 14.7 months with sorafenib (hazard ratio (HR) 0.85; *p* = 0.075); the protocol-defined significance level was not reached [12]. In KEYNOTE-240, mOS was 13.9 months for pembrolizumab versus 10.6 months for placebo (HR 0.781; *p* = 0.0238), and median progression-free survival (mPFS) for pembrolizumab was 3.0 months versus 2.8 months (HR 0.718, *p* = 0.0022) [13]. In contrast, the results of the phase III KEYNOTE-394 trial announced at the American Society of Clinical Oncology (ASCO) gastrointestinal (GI) cancers 2022 conference revealed an improvement in OS, PFS, and ORR for pembrolizumab plus best supportive care (BSC) versus placebo plus BSC after first-line treatment with sorafenib in Asian patients with advanced HCC. For patients treated with pembrolizumab, mOS was 14.6 months versus 13.0 months with placebo. Median PFS was 2.6 months for pembrolizumab versus 2.3 months for placebo. The ORR was 13.7% in the pembrolizumab group versus 1.3% in the BSC group, with a median duration of response of 23.9 months versus 5.6 months, “confirming that this agent has a positive effect as monotherapy” [14].

Other monotherapy agents targeting the PD-1/PD-L1 axis are being explored in first- and second-line treatment of advanced HCC, such as cemiplimab, camrelizumab, and tislelizumab. Preliminary phase I and II studies evaluating two PD-1 inhibitors, cemiplimab and camrelizumab, respectively, showed response rates between 15% and 20% [15,16]. Tislelizumab, a humanized IgG4 monoclonal antibody (mAb) with high affinity for PD-1, is being evaluated in the phase III RATIONALE-301 trial versus sorafenib in first-line treatment of advanced HCC [17]. The preliminary results of the phase Ia/Ib study showed an ORR of 12% (*n* = 6/50) for tislelizumab monotherapy in pretreated advanced HCC [17]. The same drug gave convincing results in the single-arm phase II RATIONALE-208 study that recruited 249 patients with advanced HCC previously treated with at least one prior line of therapy. The ORR was 13.3% and the disease control rate (DCR) was 53.0%. A response duration of at least 12 months was observed in 79.2% of patients [18].

Finally, ICI monotherapy in advanced HCC has shown disappointing results, leading to the development of combination agents including ICI (ICI plus antiangiogenic, combination of 2 ICI) [19].

## 3. ICI Combination Therapy in Advanced HCC: More Successes Than Failures

### 3.1. ICI in Combination with Antiangiogenic Therapies

Aside from ICIs, antiangiogenic therapies form the other major type of systemic treatment for HCC. This tumor type and its TME are hypoxic, with aberrant neoangiogenesis due to tumor-released proangiogenic factors, including VEGF [20]. These proangiogenic factors promote an immunosuppressed TME by preventing the recruitment of CD8+ T-cells and dendritic cells (DCs), allowing the influx of Treg cells and increasing the expression of PD-1 and PD-L1 on the surface of immune and tumor cells [7,8,21]. Antiangiogenic drugs normalize tumor vascularization, allowing tumor hypoxia, the recruitment of effector immune cells (DCs, T-cells, or neutrophils), and downregulation of immunosuppressive cells (myeloid-derived suppressor cells and Treg cells) [3,22,23,24]. This immunomodulatory effect explains their synergic effect with ICIs.

#### 3.1.1. Atezolizumab + Bevacizumab: A New Standard of Care (Table 1)

Following the practice-changing phase III IMbrave150 trial, the combination of atezolizumab (anti-PD-1 mAb) and bevacizumab (anti-VEGF mAb) was approved as a first-line treatment in advanced HCC, replacing sorafenib, thanks to an impressive mOS of 19.2 months (vs. 13.4 months for sorafenib, HR = 0.66 *p* < 0.001) [25,26,27]. The mPFS was 6.9 months with atezolizumab plus bevacizumab and 4.3 months with sorafenib (HR 0.65; *p* < 0.001); ORRs were 30% versus 11% in the respective treatment groups [26,27]. The study also revealed positive patient-reported quality-of-life outcomes, with a longer time to deterioration for patients receiving atezolizumab plus bevacizumab versus those receiving sorafenib (11.2 months vs. 3.6; HR 0.63) [28]. Since these results, the combination of atezolizumab and bevacizumab has been considered to be the new standard of care in first-line HCC and has been approved for first-line treatment of advanced HCC in more than 80 countries to date [29].

**Table 1 cancers-14-04523-t001:** Two new standards of care for advanced HCC systemic treatment: ICI-based combinations.

Study	Study Treatment	Control Arm	Number of pts	ORR (%)	mPFS (mo)	mOS (mo)
IMbrave 150 [25,26,27]	Atezolizumab Bevacizumab	Sorafenib	501 (336 in atezo–bev arm vs. 165 sor arm)	30% (vs. 11% in sor arm)	6.9 mo (vs. 4.3 mo in sor arm)	19.2 mo (vs. 13.4 mo in sor arm)
HIMALAYA [30]	Durvalumab Tremelimumab (STRIDE) (and durvalumab alone)	Sorafenib	1171 (393 in STRIDE arm, 389 in durva arm, 389 in sor arm)	20.1% with STRIDE17% with durva 5.1% with sor	3.78 with STRIDE (HR = 0.90, NS) 3.65 with durva (HR = 1.02, NS) vs. 4.07 mo with sor	16.43 mo with STRIDE (HR 0.78 *p* = 0.0035), 16.56 mo with durva (vs. 13.77 mo with sor)

Abbreviations: atezo–bev = atezolizumab–bevacizumab; sor = sorafenib; mo = months; durva = durvalumab.

#### 3.1.2. Other Combinations of ICIs and Antiangiogenic Therapy

Focusing on phase III studies, four recent or ongoing trials are evaluating the combination of antiangiogenic drugs and ICIs (Table 2). First, the phase III COSMIC-312 trial (NCT03755791), the results of which were recently published, did not meet its dual primary endpoint, with a significant improvement in PFS with cabozantinib plus atezolizumab versus sorafenib in treatment-naive HCC, but no difference in OS. The mPFS was 6.8 months with cabozantinib–atezolizumab versus 4.2 months with sorafenib (HR 0.63; *p* = 0.0012). The mOS were 15.4 months and 15.5 months in cabozantibib–atezolizumab arm and sorafenib arm, respectively. The results of this study do not support the use of cabozantib plus atezolizumab as a standard first-line treatment [31]. Second, the LEAP-002 study (NCT03713593) is evaluating the combination of pembrolizumab and lenvatinib versus lenvatinib monotherapy, based on the excellent results of the phase Ib KEYNOTE-524 trial [32], which showed an ORR of 36%, mPFS of 8.6 months, and mOS of 22 months. In a very recent press release, EISAI and Merck& Co. stated that the LEAP 002 trial failed its primary endpoint with only a trend toward improved progression-free survival and overall survival without reaching significance. Final results of the study are pending [33]. Third, a Chinese phase III study is currently evaluating camrelizumab (anti-PD-1 mAb) and apatinib (anti-VEGFR-2 mAb) against sorafenib in first-line advanced HCC (NCT03764293). The camrelizumab and apatinib combination was previously tested in the phase II RESCUE trial in both first- and second-line settings [34] and showed promising ORRs of 34.3% and 22.5%, respectively. The 12-month survival rates were 74.7% and 68.2%, respectively. Finally, the combination of sintilimab (a PD-1 inhibitor) and a biosimilar of bevacizumab (IBI305) is under evaluation versus sorafenib in advanced HCC in the phase II/III ORIENT-32 trial. The interim analysis found a significant improvement in PFS and OS (mPFS 4.6 vs. 2.8 months, HR 0.57, *p* < 0.0001; mOS not reached vs. 10.4 months, HR 0.57, *p* < 0.0001) [35]. In all trials, toxicity was manageable. There is a very high probability that one or several of these new combinations will show a significant positive effect in terms of efficacy versus sorafenib, challenging the role of the atezolizumab–bevacizumab combination. Without any direct comparison of these regimens, clinicians will choose strategies based on the reported toxicities and on some differences in inclusion criteria between the trials.

Other ICI and antiangiogenic tyrosine kinase inhibitor (TKI) combinations are currently being evaluated in advanced HCC in several phase I and II trials; the main ones are presented in Table 3.

### 3.2. Combinations of ICIs

#### 3.2.1. Durvalumab + Tremelimumab: Another Standard of Care (Table 1)

The ICI combinations typically investigated are anti-CTLA-4 antibodies combined with anti-PD-1 or anti-PD-L1 antibodies. HIMALAYA was an open-label, multicenter, global, phase III trial including patients with confirmed BCLC B (not eligible for locoregional therapy) and BCLC C HCC. The inclusion criteria were similar to those of the IMbrave150 study, except that patients with main portal vein occlusion were excluded from this study. The study included 1324 patients who were randomized to receive either sorafenib or durvalumab, or a combination of durvalumab and tremelimumab (one initial dose), or a combination of durvalumab and tremelimumab (four doses given every 4 weeks). This last arm using repeated doses of tremelimumab was closed following a preplanned analysis of a phase II study (153 patients were included in this arm) [30]. Initial clinical characteristics were well balanced between the arms; about 80% of the patients had BCLC B HCC, and just over 60% of the patients had an ECOG performance status score of 0. The cause of chronic liver disease was hepatitis B in about 31% of cases, hepatitis C in 27–28% of cases, and nonviral in 41–43% of cases. The mOS was 16.4 months in the tremelimumab and durvalumab arm and 13.8 months in the sorafenib arm (HR 0.78, *p* = 0.0035). It was 16.6 months in the durvalumab alone group. mPFS was 3.78, 3.65, and 4.07 months, with no significant differences, in the tremelimumab plus durvalumab, durvalumab alone, and sorafenib alone arms, respectively. The ORRs were 20.1% 17.0%, and 5.1%, in the tremelimumab plus durvalumab, durvalumab alone, and sorafenib alone arms, respectively. There were no major tolerance issues, particularly in the combination arm [30]. Despite the surprising lack of significant differences in terms of PFS, these positive OS results suggest that this combination of tremelimumab and durvalumab should be considered as another major option for first-line treatment of advanced HCC.

#### 3.2.2. Other ICI Combinations

Among other ICI combinations, one of the most advanced is the well-known and well-used combination of ipilimumab and nivolumab that has proven to be active and powerful in many cancers, including melanoma and renal cancer [39,40]. In the CheckMate 040 randomized phase II study, 148 patients were randomized in a three-arm study to receive either nivolumab 1 mg/kg and ipilimumab 3 mg/kg every 3 weeks four times followed by nivolumab 240 mg every 2 weeks (arm A), nivolumab 3 mg/kg and ipilimumab 1 mg/kg every 3 weeks four times followed by nivolumab 240 mg every two weeks (arm B), or nivolumab 3 mg/kg every 2 weeks and ipilimumab 1 mg/kg every 6 weeks (arm C) [41]. Response rates were approximately 30% in each arm, and considering the safety profile in this and other studies, the combinations of nivolumab 3 mg/kg and ipilimumab 1 mg/kg were chosen for the ongoing phase III CheckMate 9DW, with sorafenib as a control group (NCT04039607).

## 4. Combinations of ICI and Chemotherapy in Advanced HCC

The oxaliplatin plus 5-fluorouracil-based chemotherapy regimen (FOLFOX4) is approved in China for the treatment of advanced HCC. Approval was based on the results of the EACH trial, which compared FOLFOX and doxorubicin in Asian patients with advanced HCC (DCR: 47%; mOS: 5.7 months) [42]. It has been reported that oxaliplatin can induce immunogenic cell death, inducing release of danger signals or damage-associated molecular patterns and recruitment of CD8+ T-cells [43]. A Chinese phase II study of camrelizumab combined with oxaliplatin-based chemotherapy (FOLFOX or gemcitabine plus oxaliplatin (GEMOX)) in patients with advanced pretreated HCC reported that among the 34 evaluable HCC patients, confirmed ORR was 29.4% and DCR 79.4%. mPFS and mOS were 7.4 and 11.7 months. Grade ≥ 3 immune-related adverse events occurred only in 5.9% of patients (NCT03092895) [44,45]. A phase III trial is currently underway to confirm the efficacy and safety of camrelizumab plus FOLFOX4 versus placebo plus FOLFOX4 for the first-line treatment of advanced HCC (NCT03605706).

## 5. Other ICIs in Advanced HCC

Another therapeutic avenue being investigated concerns anti-LAG3 (lymphocyte-activation gene 3) and anti-TIM3 (T-cell immunoglobulin mucin family member 3) antibodies. TIM3 and LAG3 are less well-known immune checkpoints. They are expressed on tumor-infiltrating lymphocytes (TILs) in HCC, downregulate T-cell activation, and participate in T-cell exhaustion via various mechanisms. TIM3 inhibitors are under evaluation in combination with anti-PD-1 antibodies in still-recruiting phase II trials (NCT03680508, NCT03744468). This combination has shown interesting response rates in other tumor types [46]. Isolated LAG3 blockade showed very limited anti-tumor activity, while combined with other ICIs such as PD-1 inhibitors, it offers promising results. In this context, the phase II study RELATIVITY-073 is exploring the combination of relatlimab (the first-in-class anti-LAG3 mAb) with nivolumab in patients with advanced HCC only pretreated with TKIs and no prior immune therapy (NCT04567615). Bispecific anti-LAG3/anti-PD-1 antibodies are currently under investigation, including in the field of advanced HCC in phase I and phase II studies (NCT04524871, NCT04140500, NCT03440437).

### Other New Strategies: The Morpheus Liver Trial as an Example

New designs of platform trials allow evaluation of new strategies and sometimes combinations of these new strategies. For instance, cohort 1 of the Morpheus phase Ib/II trial is exploring several innovative drugs in combination with the approved atezolizumab–bevacizumab combination, namely, RO7247669 (an anti-PD-1/LAG3 bispecific antibody), tiragolumab (an anti-TIGIT antibody), tocilizumab (an anti-IL-6 receptor antibody), SAR439459 (a pan-TGFβ inhibitor), or TPST-1120 (a first-in-class, oral selective PPARα antagonist), to enhance the anti-tumor immune response (NCT04524871).

## 6. ICI Combined with Local Therapy in Early HCC

Although immunotherapy is only approved for the treatment of advanced HCC, several trials are exploring its application in earlier stages of the disease. To date, no adjuvant treatment has been shown to be effective after ablative treatment of HCC. The first positive data from trials of immunotherapy in advanced HCC, plus its good clinical tolerance, raise interest in its potential effectiveness as an adjuvant treatment.

### 6.1. Adjuvant ICI with a Curative Intent

At least five ongoing phase III trials are investigating anti-PD-1 or anti-PD-L1 mAbs (pembrolizumab, nivolumab, durvalumab, atezolizumab, and toripalimab) with or without antiangiogenic agents as adjuvant therapy after surgery or thermic local ablation for localized HCC (Table 4). These large trials aimed to include between 500 and 1000 patients and had disease-free survival as the primary endpoint.

Remarkably, the addition of ICI-based neoadjuvant therapy and adjuvant therapy can lead to a pathological complete response. A phase II study evaluating nivolumab with or without ipilimumab in patients with resectable HCC revealed a pathological complete response rate of 29% (4/14), with complete response observed in two cases each in the nivolumab monotherapy and nivolumab plus ipilimumab groups [47]. Another phase II study, still recruiting, has as its primary objective assessment of the clinical activity of neoadjuvant cemiplimab therapy in several tumor types, including resectable HCC, measured by performing pathological analysis on resected tumors (NCT03916627). The first results shared at the ASCO GI cancers 2022 meeting showed that four of the 20 included patients (20%) had more than 70% tumor necrosis, including three with 100% necrosis. CT scan evaluation also revealed three (15%) partial responses. Outcomes of the ongoing adjuvant phase of cemiplimab treatment of this same trial are pending.

Locoregional treatments are used in patients with early-stage HCC (BCLC 0-A and B). The main modalities are thermic ablation methods such as radiofrequency ablation (RFA), cryoablation, and stereotactic body radiation therapy (SBRT) [29]. These tumor-destructive treatments lead to the release of neo-antigens which could enhance the response to immunotherapy [48].

Currently, several phase III trials are exploring immunotherapy as an adjuvant to local ablative therapy. Among them are trials mentioned above concerning both operated and locally treated patients (see Table 4). In addition, the NIVOLEP phase II trial (NCT03630640) is evaluating nivolumab in neoadjuvant and adjuvant settings in patients with advanced HCC treated by electroporation with a curative intent. Electroporation is a new percutaneous ablation approach that induces tumor cell apoptosis using a pulse of electricity to open the pores in cell membranes.

### 6.2. ICI Combined with SBRT

The combination of ICI with SBRT is still under investigation and is supported by the concept of the abscopal effect and the synergistic effect of these two treatments already observed in several solid tumors [49]. The phase II/III trial ISBRT01 (NCT04167293) aims to investigate efficacy and safety of SBRT followed by sintilimab (an anti-PD-1 mAb) compared with SBRT alone for patients with HCC with portal vein invasion who have received arterially directed therapy. An ongoing phase II study (NCT03316872) is exploring the efficacy of the combination of pembrolizumab and SBRT in patients with advanced HCC who have experienced disease progression after treatment with sorafenib.

### 6.3. ICI combined with Palliative Local Therapy

Trans-arterial chemoembolization (TACE) and trans-arterial radioembolization (TARE) are used in patients with intermediate-stage HCC (BCLC B) [29].

#### 6.3.1. TACE plus ICI

A significant proportion of patients with HCC present with intermediate-stage disease and these patients are typically treated with TACE. However, long-term survival remains short after TACE treatment, with many patients relapsing in the first year post-treatment. This calls for the development of strategies to extend the duration of response. TACE may liberate an abundance of tumor antigens and proinflammatory cytokines, which supports an approach combining it with immunotherapy drugs. In a phase II study exploring peri-interventional tremelimumab therapy in 32 patients treated with subtotal RFA (*n* = 12) or chemoablation (*n* = 11) for advanced HCC (25% BCLC B; 75% BCLC C), 26% of evaluable patients had a confirmed partial response, and the mOS was 12.3 months (NCT01853618) [50]. This trial was encouraging in terms of the efficacy and safety of ICIs combined with TACE/RFA in the treatment of advanced HCC [50]. In addition, results of a phase I clinical trial evaluating nivolumab combined with drug-eluting bead TACE (DEB-TACE) for the treatment of HCC, which included nine patients with BCLC stage B and Child–Pugh grade A disease, showed that two of the nine evaluable patients achieved a partial response and two had stable disease. The 12-month OS was 71% [51]. Five ongoing phase III trials aim to evaluate the combination of TACE and anti-PD-1/PD-L1 antibodies with or without anti-CTLA-4 therapy and other targeted therapies (TKIs or antiangiogenic mAbs) (Table 5).

#### 6.3.2. ICI Therapies via Intra-Arterial Infusion or Intratumoral Routes

Another therapeutic strategy under development consists of the administration of immunotherapy drugs by hepatic arterial infusion or directly into a tumor site, allowing the concentration of the drug to be increased at the target while sparing the healthy parenchyma. A phase III Chinese trial was designed to investigate survival outcomes, response rates, and safety of patients with advanced HCC (BCLC C stage) treated with PD-1/PD-L1 inhibitors by hepatic artery versus vein infusion (NCT03949231). Another phase III Chinese trial is exploring the same outcomes for trans-arterial/intratumoral infusion of anti-PD-1/PD-L1 antibodies and/or the anti-CTLA-4 antibody ipilimumab for several tumor types and includes an advanced HCC cohort (NCT03755739). Other early phase trials are exploring these routes of administration, such as the phase I HIPANIV trial (Hepatic Intra-Arterial Administration of Ipilimumab in Combination with Intra-venous Nivolumab for Advanced Hepatocellular Carcinoma, NCT04823403).

#### 6.3.3. Selective Internal Radiation Therapy/TARE + ICI

Similar to other locoregional treatments, hepatocyte exposure to yttrium-90 microspheres via TARE (Y-90 TARE) results in immunogenic cell death and tumor-specific immunity, suggesting that this treatment may synergize with checkpoint inhibitor therapies to improve response rates and disease control. Three phase I or II studies evaluating the combination of Y-90 TARE with nivolumab found response rates of approximately 30% and/or DCRs of 80% (Table 6) [54,55,56].

Two other ongoing phase I studies, for which results are pending, are investigating the combination of anti-PD-1 with Y-90 TARE (nivolumab: NCT03812562; pembrolizumab: NCT03099564). There are currently no phase III trials in this indication; further explorations are needed.

#### 6.3.4. Local Ablation to Reboot Sensitivity to ICI

An interesting strategy would be to enhance the anti-tumor immunity with local ablation and thus reboot sensitivity to ICIs. A phase II trial (NCT03939975) assessed the response of 50 patients with advanced HCC for whom first-line treatment with sorafenib had failed and who had started second-line treatment with anti-PD-1 therapy (pembrolizumab or nivolumab). Of the 50 patients treated, the rates of response, stable disease, and atypical and typical progression were 10% (*n* = 5), 42% (*n* = 21), 32% (*n* = 16), and 12% (*n* = 6), respectively. Thirty-three patients who had an atypical response to anti-PD-1 inhibitors or who had stable disease also underwent subtotal thermal ablation. Additional ablation improved efficacy with tolerable toxicity, and the ORR increased from 10 to 24% (12/50). The mPFS and mOS were 5.0 and 16.9 months, respectively [57].

## 7. New Strategies

Aside from ICIs, new therapeutic approaches target tumor-specific antigens (TSA: antigens only expressed by tumor cells) and tumor-associated antigens (TAA: antigens overexpressed by tumor cells but also expressed by healthy cells) such as glypican-3 (GPC3) or alpha-fetoprotein (AFP).

### 7.1. Cell Therapies (CT)

ACT consists of the activation and/or proliferation of autologous leukocytes by exposure to tumor antigens and/or specific cytokines ex vivo and their reinjection into the patient. Different trials are underway in HCC with chimeric antigen receptor (CAR) T-cells, cytokine-induced killer cells (CIK), and TILs.

One of the most promising CT strategies in HCC appears to be CAR T-cell therapy. CAR T-cells are not MHC-restricted; they are engineered to bind to specific TAA and enhance immune responses against tumor cells. GPC3, overexpressed in more than 70% of HCC, is the most specific and most extensively studied TAA in the production of CAR T-cells for HCC. The results of a phase I study exploring fourth-generation CAR T-cells targeting GPC3 (4G-CAR-GPC3 T-cells) for advanced HCC were shared at the ASCO GI cancers 2021 annual meeting. In this single-arm, open-label, first-in-human phase I trial (NCT03980288), Fang and colleagues investigated the safety and anti-tumor activity of autologous 4G-CAR-GPC3 T-cells for patients with GPC3+ heavily pretreated advanced HCC. Six subjects with HBV-related metastatic HCC were enrolled. All patients developed grade 2 or 3 cytokine release syndrome (CRS), which was managed with tocilizumab and corticosteroids. The ORR and DCR were 16.7% (*n* = 1/6) and 50% (*n* = 3/6), respectively. The mPFS was 4.2 months [58]. Joint analysis of two phase I studies exploring second-generation CAR-GPC3 T-cells revealed a DCR of 23% (*n* = 3/13) and an mOS of 6 months. Seventy percent of patients developed CRS; one death occurred [59]. Currently, six recruiting phase I studies continue to investigate CAR-GPC3 T-cells (NCT02905188, NCT03884751, NCT05003895, NCT04121273, NCT05103631, and NCT03198546).

Many single-arm or small controlled trials have suggested that CIK cells have activity against HCC, both in adjuvant settings and at advanced stages. CIK cells are a type of anti-tumor T-cells characterized by the co-expression of the T-cell marker CD3 and the NK-cell marker CD56 molecules, which can easily be generated by expanding human peripheral blood mononuclear cells in the presence of interferon-_γ_ (IFN-_γ_), anti-CD3 antibodies, and IL-2 [60]. A phase I basket trial of CIKs for advanced stage, refractory renal cell carcinoma, HCC, or lymphoma treated 12 patients and reported a DCR of 42% [61]. A retrospective study evaluated the efficacy of autologous CIK cell transfusion in combination with TACE and RFA versus sequential therapy with TACE and RFA in HCC. Patients in the TACE/RFA/CIK group had significantly longer OS (56 vs. 31 months, *p* = 0.001) and PFS (17 vs. 10 months, *p* = 0.001) than those in the TACE/RFA group [62].

Two meta-analyses of clinical trials evaluating “adjuvant” CIKs in BCLC A/B disease after surgery, local ablation, or TACE suggest that CIK cells may reduce tumor recurrence rates [63,64]. In a phase III trial (NCT00699816) of 230 patients with HCC who underwent curative treatment (surgical resection, RFA, or percutaneous ethanol injection), adjuvant immunotherapy with activated CIK cells (CD3+/CD56+ and CD3+/CD56− T-cells and CD3−/CD56+ NK cells) increased recurrence-free survival (44 vs. 30 months) [65]. Two other phase III studies are exploring CIK cells as adjuvant therapy of resected HCC (NCT01749865) or in combination with TACE (NCT03592706); results are pending.

TILs are polyclonal, tumor-targeting T-cells expanded from patient tumor biopsies ex vivo for use as autologous therapeutics. The feasibility of adjuvant TIL therapy was shown in a phase I trial in patients with HCC (NCT04538313). Further data are needed to determine the activity of TILs in HCC.

### 7.2. Oncolytic Viruses and Vaccines

Oncolytic viruses specifically infect tumor cells. As they multiply inside cancer cells, the weakly immunogenic tumor cells allow the intracellular viruses to grow exponentially, resulting in cell lysis. Pexastimogene devacirepvec (Pexa-Vec or JX-594) is engineered to preferentially replicate in and destroy tumor cells while stimulating anti-tumor immunity by its expression of GM-CSF. Results from a phase IIa study in which 30 patients received intratumoral injections of JX-594 performed by interventional radiologists showed a statistically significantly improved OS for patients with advanced liver cancer who received a high dose of Pexa-Vec compared with the group receiving the low dose—14.1 months versus 6.7 months [66]. However, subsequent phase IIb (TRAVERSE) and phase III (PHOCUS) studies failed to show a survival benefit of JX-594 plus sorafenib versus sorafenib alone [67,68].

Several phase I studies are exploring peptide vaccines targeting TSAs and TAAs, in particular AFP (NCT00005629, NCT00022334), NY-ESO-1 (NCT01522820), and p53 (NCT02432963). The injection of these peptide vaccines alone or in combination with an ICI aim to induce an immune response directed against cells expressing the targeted tumor antigen. Another vaccine strategy in advanced HCC consists of a DC vaccine (NCT01974661 [69], NCT04912765, NCT04147078) in combination with other therapies (such as TACE, sorafenib, or nivolumab). Monocyte-derived allogeneic DCs are stimulated with a combination of proinflammatory factors. These can be administered subcutaneously or via intratumoral injections, which could induce recruitment of immune cells, including NK cells, DCs, and T-cells to the injection site, leading to local tumor cell killing and release of cell-associated tumor antigens. More explorations are needed to demonstrate the efficacy of these innovative strategies.

## 8. Conclusions

The atezolizumab–bevacizumab combination has become a standard of care as first-line treatment in advanced HCC since the results of the IMbrave 150 trial. In the same indication, the durvalumab–tremelimumab combination has been added to the therapeutic armamentarium since the very recent results of the Himalaya trial. The latest and recent version of the BCLC now includes these two front-line options for advanced stages (BCLC C) [70]. Currently, several phase III trials evaluating combinations of ICI and antiangiogenic agents have yielded results still insufficient to compete with atezo–bevacizumab or durvalumab–tremelimumab combinations. For the earlier stages of HCC (BCLC 0-A and B), trials combining immunotherapy with locoregional treatments are underway. The idea is attractive, as the combination of immunotherapy and locoregional treatment has given very interesting results in other indications [71,72,73,74]. In this indication there are promising results from phase I/II trials of ICIs in combination with RFA, TACE, Y-90 TARE, or SBRT, and several phase III trials are ongoing.

New strategies are being investigated in the field of adoptive cell therapies (CAR T-cells, CIK cells, TILs), both in early/intermediate disease stages in combination with local treatment and in advanced stages after the failure of standard treatments. Phase III results will be needed to draw more conclusions.

Overall, immunotherapy definitely has a place in the management of advanced HCC and probably at an earlier stage in combination with local therapies, but results from phase III trials are needed to confirm this.

## Figures and Tables

**Table 2 cancers-14-04523-t002:** Available results of phase III studies evaluating ICI in combination with antiangiogenic agents in advanced HCC.

Study	Study Treatment	Control Arm	Number of pts	ORR (%)	mPFS (mo)	mOS (mo)
IMbrave 150 [25,26,27]	Atezolizumab Bevacizumab	Sorafenib	501 (336 in atezo–bev arm vs. 165 sor arm)	30% (vs. 11% in sor arm)	6.9 mo (vs. 4.3 mo in sor arm)	19.2 mo (vs. 13.4 mo in sor arm)
COSMIC-312 [31]	Atezolizumab Cabozantinib	Sorafenib (and cabozantinib alone for safety explorations)	837 (432 in atezo–cabo arm, 217 in sor arm, 188 in cabo arm)	11% (vs. 4% in sor arm)	6.8 mo (vs. 4.2 mo in sor arm)	15.4 mo (vs. 15.5 mo in sor arm)
ORIENT-32 [35]	Sintilimab–bevacizumab biosimilar	Sorafenib	571 (380 in sinti-bev biosim. arm; 191 in sor arm)		4.6 mo (vs. 2.8 mo in sor arm)	NR (vs. 10.4 mo in sor arm)

Abbreviations: atezo–bev = atezolizumab–bevacizumab; sor = sorafenib; mo = months; atezo–cabo = atezolizumab–cabozantinib; sinti-bev biosimilar = sintilimab–bevacizumab biosimilar.

**Table 3 cancers-14-04523-t003:** Phase I/II studies of immune checkpoint inhibitors and antiangiogenic tyrosine kinase inhibitors in advanced hepatocellular carcinoma.

NCT/Reference	Drug Combination	Phase	Number of pts	ORR (%)
NCT03418922 [36]	Nivolumab–lenvatinib	Phase Ib	30	77
NCT03439891	Nivolumab–sorafenib	Phase II	24 (still recruiting)	Pending
NCT03347292 [37]	Pembrolizumab–regorafenib	Phase Ib	29	30
NCT03289533 [38]	Avelumab–axitinib	Phase Ib	22	30
NCT02988440	Spartalizumab–sorafenib	Phase Ib	20	Pending

Abbreviations: ORR: objective response rate.

**Table 4 cancers-14-04523-t004:** Phase III trials evaluating immunotherapy as adjuvant therapy after surgery or local ablation for localized hepatocellular carcinoma.

Identifier	ICI +/− Other Drug	Target	Study Title	*n*	Primary Outcome	Status
NCT03867084	Pembrolizumabvs. placebo	PD-1	Safety and efficacy of pembrolizumab vs. placebo as adjuvant therapy in participants with HCC and complete radiological response after surgical resection or local ablation (MK-3475-937/KEYNOTE-937)	950	RFS/OS	Recruiting
NCT03383458	Nivolumab vs. placebo	PD-1	A study of nivolumab in participants with HCC who are at high risk of recurrence after curative hepatic resection or ablation (CheckMate 9DX)	530	RFS	Active Not recruiting
NCT03847428	Durvalumab +/− bevacizumab vs. placebo	PD-L1	Assess efficacy and safety of durvalumab alone or combined with bevacizumab in patients with HCC at high risk of recurrence after curative treatment (EMERALD-2)	888	RFS	Active Not recruiting
NCT04102098	Atezolizumab + bevacizumab vs. no therapy	PD-L1	Study of atezolizumab plus bevacizumab vs. active surveillance as adjuvant therapy in patients with HCC at high risk of recurrence after surgical resection or ablation (IMbrave050)	662	RFS	Active Not recruiting
NCT03859128	Toripalimab vs. placebo	PD-1	Toripalimab or placebo as adjuvant therapy in HCC after curative hepatic resection (JUPITER 04)	530	RFS	Active Not recruiting

Abbreviations: HCC: hepatocellular carcinoma; OS: overall survival; RFS: relapse-free survival.

**Table 5 cancers-14-04523-t005:** Current phase III trials of immune checkpoint inhibitors in combination with trans-arterial chemoembolization.

Identifier	Study Title	Arms	Targets	Primary Outcome	Status
NCT04229355	DEB-TACE plus Lenvatinib or Sorafenib or PD-1 Inhibitor for Unresectable Hepatocellular Carcinoma	DEB-TACE + sorafenib vs. DEB-TACE + lenvatinib vs. DEB-TACE plus PD-1 inhibitor	PD-1	PFS	Recruiting
NCT04246177	Safety and efficacy of Lenvatinib with Pembrolizumab in combination with TACE in participants with incurable/nonmetastatic HCC (LEAP-012) [52]	TACE + lenvatinib + pembrolizumab vs. TACE + double placebo	PD-1	PFS/OS	Recruiting
NCT04268888	Nivolumab in combination with TACE/TAE for patients with intermediate stage HCC	TACE + nivolumab vs. TACE + placebo	PD-1	OS/TTTP	Recruiting
NCT03778957	A global study to evaluate TACE in combination with durvalumab and bevacizumab therapy in patients with locoregional HCC (EMERALD-1) [53]	TACE + durvalumab + bevacizumab vs. TACE + double placebo	PD-L1	PFS	Active, not recruiting
NCT04340193	A study of nivolumab and ipilimumab in combination with TACE in participants with intermediate-stage liver cancer (CheckMate 74W)	TACE + nivolumab +/− ipilimumab vs. TACE + double placebo	PD-1 CTLA-4	OS/TTTP	Active, not recruiting

Abbreviations: DEB: drug-eluting beads; HCC: hepatocellular carcinoma; OS: overall survival; TACE: trans-arterial chemoembolization; TTTP: time to TACE progression.

**Table 6 cancers-14-04523-t006:** Completed trials of immune checkpoint inhibitors in combination with Y-90 trans-arterial radioembolization.

NCT/Reference	Drug Combination	Phase	Number of pts	ORR (%)	DCR (%)
NCT02837029 [54]	Y-90 TARE + nivolumab	Phase I	11	-	82%
NCT03380130 NASIR-HCC [55]	Y-90 TARE + nivolumab	Phase II	41	38%	81%
NCT03033446 [56]	Y-90 TARE + nivolumab	Phase II	40	30%	-

Abbreviations: DCR: disease control rate; ORR: objective response rate; TARE: trans-arterial radioembolization; Y-90: yttrium-90.

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
