# Peer review of "Immunotherapy and Hepatocellular Cancer: Where Are We Now?"

_cancers, 2022, doi:10.3390/cancers14184523_

Round 1

Reviewer 1 Report

Immune checkpoint inhibitors (ICIs) including pembrolizumab, nivolumab, durvalumab, atezolizumab, etc. have been recently evaluated in HCC patients, and clinical trials assessing single-agent ICI have reported disappointing results. Conversely, immune-based combinations have been more striking. In fact, the phase III IMbrave150 trial assessing the combination of the antiangiogenic agent bevacizumab plus the PD-L1 inhibitor atezolizumab versus single-agent sorafenib has established a new standard of care for HCC patients with advanced disease. According to IMbrave150, atezolizumab - bevacizumab have reported statistically significant and clinically meaningful benefits in several clinical outcomes, including objective response rate (ORR), progression-free survival (PFS), and overall survival (OS), with these advantages also confirmed by the updated results of this trial, showing a median OS of more than 19 months in HCC patients receiving the immune-based combination. Despite ICI seem to have finally found their role in HCC as part of combinatorial strategies, several questions remain unanswered. Among these, the lack of validated biomarkers of response represents an important issue since only a proportion of HCC patients benefit from immunotherapy. Based on these premises, a greater understanding of the role of potential biomarkers including programmed death ligand 1 (PD-L1) expression, tumor mutational burden (TMB), microsatellite instability (MSI) status, gut microbiota and several others is fundamental. In addition, clinical trials on HCC immunotherapy widely differed in terms of drugs, patients, designs, terms of study phases, and inconsistent clinical outcomes.
Based on these premises, the review assesses a current, timely topic.
We recommend some changes:
- We believe this article is suitable for publication in the journal although some revisions are needed. In particular, COSMIC-312 trial has been published, and the references should be updated. In addition, several Hazard Ratios and clinical outcomes are not updated (for example, those from IMbrave150 phase III trial). Thus, the authors should update and add several clinical results - also the updated results presented by Finn of IMbrave150.

- In addition, a table summarizing the main clinical results provided by immune-based combinations as first-line treatment should be included.

- A table summarizing the synergistic action of antiangiogenic agents plus IO should be included.

- The background of the changing scenario of medical treatment in HCC should be better discussed in the introduction section, and some recent papers regarding this topic should be included, only for a matter of consistency (PMID: 34431725 ; PMID: 29968763  ).
Some changes are necessary.

Author Response

Dear Editor and Reviewers,

Thank you very much for the consideration given to our work.

Coauthors and I very much appreciated the encouraging and constructive comments on the manuscript made by the reviewers.

Regarding the comments of reviewer 1 :

1.We believe this article is suitable for publication in the journal although some revisions are needed. In particular, COSMIC-312 trial has been published, and the references should be updated. In addition, several Hazard Ratios and clinical outcomes are not updated (for example, those from IMbrave150 phase III trial). Thus, the authors should update and add several clinical results - also the updated results presented by Finn of IMbrave150.

Thanks a lot for these remarks and suggestions. We up-dated the reference and results for COSMIC-312 trial and IMbrave-150.

  1. In addition, a table summarizing the main clinical results provided by immune-based combinations as first-line treatment should be included.

Thank you very much for the suggestion , we added this table.

  1. A table summarizing the synergistic action of antiangiogenic agents plus IO should be included.

Thank you very much for the suggestion , we added this table.

4.The background of the changing scenario of medical treatment in HCC should be better discussed in the introduction section, and some recent papers regarding this topic should be included, only for a matter of consistency (PMID: 34431725 ; PMID: 29968763  ).

Thanks a lot for this interesting suggestion, we have chosen to add the first reference (PMID: 34431725) but not the second one (PMID: 29968763) not focusing on immunotherapy in HCC and being difficult to relate to our work.

We hope you will find this revised manuscript convenient for publication.

We thank you for your kind consideration and look forward to your reply.

Sincerely,

Marine VALERY

Reviewer 2 Report

This is a well written review by outstanding experts in the field of clinical immunotherapy in hepatocellular cancer. They have ample experience and a track record in this regard and have been involved in landmark trials such as IMbrave150, establishing Atezolizumab plus Bevacizumab for unresectable hepatocellular carcinoma as the current new treatment standard.

In this regard I have no doubt the authors are unquestionably beyond reproach.

The manuscript provides somewhat of a tour de force through the current state of the art in immune checkpoint therapy in (advanced) HCC, laying out treatment combinations and providing orientation and structure in a complex field. Further and importantly relevant questions arising in the context of these new therapies are raised.

This is commendable work and a great summary, although with some lengths owing to the topic. The amount of data summarized and within their scope is impressive and this is high quality work and provides a good outline for the interested reader. Further, the included tables are clearly of added value.

I only have a few minor issues and discretionary remarks:

1.) In the Title as well as in the abstract the authors ask 4 (If I am counting right?) different and very relevant questions. Ultimately none of them is resumed within in the manuscript or even attempted to be answered. Do the authors do this on purpose?

The authors refer to “two main questions” in the abstract but pose more questions. Further the included questions are of the not formulated as such or marked by a “?” This is confusing to mw.

2.) The authors mention this to be a position paper only once in the end of the abstract. This fact – if relevant- does not seem to stand out too much. Maybe the authors would like to emphasize this fact more and in case this is intended whom they represent in this coherence.

3.) Headings:
a.) 3. ICI combination therapy in advanced HCC: more successes than failures (13–15): What does (13-15) mean here?
b.) 4. Combinations of ICI and chemotherapy: Consider adding “in advanced HCC”
c.) 5. Other ICIs in HCC: Which HCCs are concerned here. Maybe include?
d.) p. 8 below Table/ first paragraph: Maybe consider including a new header here.
e.) 7.1 Adoptive cell therapy (ACT): This header seems incorrect since this is a bunch of cellular therapies not only ACT

4.) Why do the authors omit/ miss to mention the current 2022 BCLC update (Reig et al. J Hepatol. 2022 Mar;76(3):681-693.), which highlights immunotherapy in the treatment of advanced stage HCC? Would it make sense to mention and contextualize this somewhere?

5.) In their very short “introduction” the authors refer to the role of the liver and to immunosuppressive mechanisms. Further I appreciate the inclusion of modes of action, where known. However, the authors disregard the mechanisms that underlie cellular immunity in HCC, which is doubtlessly essential to enable recognition and T cell-mediated immune responses required in this context and a prerequisite for immune checkpoint inhibitors to work. This might be worthwhile to be shortly explained for the readers, even if a primarily clinical audience is intended.

Discretionary remarks:

p.1: (Introduction): „indoleamine 2,3-dioxygénase“
p.1: Treg are usually not primarily characterized by CD4+ (which is characteristic for many other cells also) but by Foxp3.p.4: 3.2.1 (line 4): I suggest „BCLC C HCC“
p.9: Is this correct: „interferon-C“?
p.9: „MHC nonspecific“ does not seem the appropriate term? Rather not MHC restricted, not MHC specific.
p.10: What is meant by „oncolytic viruses bind tumor cells“?
p.10: Switching between Pexa-Vec and JX-594 without introducing the latter is confusing to the non-expert reader.

Author Response

Dear Editor and Reviewers,

Thank you very much for the consideration given to our work.

Coauthors and I very much appreciated the encouraging and constructive comments on the manuscript made by the reviewers.

Regarding the comments of reviewer 2 :

  1. In the Title as well as in the abstract the authors ask 4 (If I am counting right?) different and very relevant questions. Ultimately none of them is resumed within in the manuscript or even attempted to be answered. Do the authors do this on purpose?

Thank you very much for this constructive comment. The conclusion has been rewritten to hopefully better answer the main question "immunotherapy in HCC: where are we now?"

We have also reworded the introduction to better connect it to the rest of the article and show that we are answering the question by presenting the results of the studies.

The authors refer to “two main questions” in the abstract but pose more questions. Further the included questions are of the not formulated as such or marked by a “?” This is confusing to me.

Thank you very much for this remark. We rephrase theses question, both introduced by “This raises the question” or “The other major question” to make it clearer.

2.) The authors mention this to be a position paper only once in the end of the abstract. This fact – if relevant- does not seem to stand out too much. Maybe the authors would like to emphasize this fact more and in case this is intended whom they represent in this coherence.

Thank you very much for this remark. We have reworded this part of the abstract for more consistency.

3.) Headings:
a.) 3. ICI combination therapy in advanced HCC: more successes than failures (13–15): What does (13-15) mean here?

Thank you very much for this suggestion. We withdraw the bibliographic references included in this title.

b.) 4. Combinations of ICI and chemotherapy: Consider adding “in advanced HCC”

Thank you very much for this valuable suggestion. We added it.

c.) 5. Other ICIs in HCC: Which HCCs are concerned here. Maybe include?

Thank you very much for this valuable suggestion. We added it.

d.) p. 8 below Table/ first paragraph: Maybe consider including a new header here.

Thank you very much for this valuable suggestion. We added it.

e.) 7.1 Adoptive cell therapy (ACT): This header seems incorrect since this is a bunch of cellular therapies not only ACT

Thank you very much for this remark. We corrected it.

4.) Why do the authors omit/ miss to mention the current 2022 BCLC update (Reig et al. J Hepatol. 2022 Mar;76(3):681-693.), which highlights immunotherapy in the treatment of advanced stage HCC? Would it make sense to mention and contextualize this somewhere?

Thank you very much for this valuable suggestion. We added the refence in our conclusion.

5.) In their very short “introduction” the authors refer to the role of the liver and to immunosuppressive mechanisms. Further I appreciate the inclusion of modes of action, where known. However, the authors disregard the mechanisms that underlie cellular immunity in HCC, which is doubtlessly essential to enable recognition and T cell-mediated immune responses required in this context and a prerequisite for immune checkpoint inhibitors to work. This might be worthwhile to be shortly explained for the readers, even if a primarily clinical audience is intended.

Thank you very much for this remark. We have improved the part on the rationale of immunotherapy in the treatment of HCC.

Discretionary remarks:

p.1: (Introduction): „indoleamine 2,3-dioxygénase“
p.1: Treg are usually not primarily characterized by CD4+ (which is characteristic for many other cells also) but by Foxp3.p.4: 3.2.1 (line 4): I suggest „BCLC C HCC“
p.9: Is this correct: „interferon-C“?
p.9: „MHC nonspecific“ does not seem the appropriate term? Rather not MHC restricted, not MHC specific.
p.10: What is meant by „oncolytic viruses bind tumor cells“?
p.10: Switching between Pexa-Vec and JX-594 without introducing the latter is confusing to the non-expert reader.

Thank you very much for these discretionary remarks. We took them into account and incorporated them into the manuscript.

We hope you will find this revised manuscript convenient for publication. We thank you for your kind consideration and look forward to your reply.

Sincerely,

Marine VALERY

Round 2

Reviewer 1 Report

acceptance.